# *Glycine soja* Leaf and Stem Extract Ameliorates Atopic Dermatitis-like Skin Inflammation by Inhibiting JAK/STAT Signaling

**DOI:** 10.3390/ijms26104560

**Published:** 2025-05-09

**Authors:** Yoon-Young Sung, Misun Kim, Dong-Seon Kim, Eunjung Son

**Affiliations:** KM Science Research Division, Korea Institute of Oriental Medicine, Daejeon 34054, Republic of Korea; kms@wilaboratory.com (M.K.); dskim@kiom.re.kr (D.-S.K.); ejson@kiom.re.kr (E.S.)

**Keywords:** *Glycine soja*, skin inflammation, skin barrier, human keratinocyte cells (HaCaT), transepidermal water loss

## Abstract

Wild soybean (*Glycine soja*, GS) is a traditional medicine used to treat inflammation. In this study, the anti-atopic properties of GS leaf and stem extract on skin inflammation were evaluated in the *Dermatophagoides farinae*-extract-induced mouse model and keratinocytes. Oral administration of the GS extract reduced scratching, dermatitis score, transepidermal water loss, thickness of epidermis, inflammatory cell accumulation, and serum concentrations of thymic stromal lymphopoietin and immunoglobulin E. GS downregulated the expression of inflammatory gene markers of atopic dermatitis (AD), including interleukin (IL)-6; regulated on activation, normal T cell expressed and secreted (RANTES); thymus- and activation-regulated chemokine (TARC); and macrophage-derived chemokine (MDC) and upregulated the expression of filaggrin, a keratinocyte differentiation marker, in skin tissue. GS downregulated Janus kinase 1, signal transducer and activation of transcription (STAT) 1, and STAT3 pathways. Using ultra-performance liquid chromatography, we identified seven flavonoids in GS extract, including apigenin, epicatechin, genistein, genistin, daidzin, daidzein, and soyasaponin Bb. GS, apigenin, and genistein reduced the expression of IL-6, MDC, TARC, and RANTES and increased filaggrin via the downregulation of STAT3 phosphorylation in interferon-γ/tumor necrosis factor-α-stimulated keratinocytes. Our results suggest that GS leaf and stem extract ameliorates AD-like skin inflammation by regulating the immune response and restoring skin barrier function.

## 1. Introduction

Atopic dermatitis (AD) is a chronic immune-mediated inflammatory skin illness characterized by itching, xerosis (skin roughness), and ichthyosis (skin keratinization disorder) [1]. AD may be caused by immunological triggers, genetic factors, and environmental factors, such as diet, microbiota, and irritants [2,3], and the pathological progression may involve epidermal barrier dysfunction and immune dysregulation [4]. Upon disruption of the epidermal barrier by defective skin barrier components, such as filaggrin, loricrin, and ceramides, keratinocytes secrete chemokines, including thymus- and activation-regulated chemokine (TARC); regulated on activation, normal T cell expressed and secreted (RANTES) chemokine; and macrophage-derived chemokine (MDC), which recruit dendritic cells, T cells, and macrophages to the inflammatory zones [5]. The release of T helper 2 (Th2) cytokines, such as thymic stromal lymphopoietin (TSLP), interleukin (IL)-4, IL-5, IL-13, and IL-31, contributes to skin inflammation, pruritus, neuromodulation of peripheral nerves associated with pruritus transduction, and regulation of the skin barrier in AD. The Janus kinase (JAK)–signal transducer and activator of transcription (STAT) signaling pathway plays an essential role in the dysregulation of immune response in AD, including exaggeration of Th2 cell responses, eosinophils activation, and modulation of the epidermal barrier and peripheral nerves involved in itch, through mainly Th2 cytokine–receptor binding and phosphorylation/activation of JAK/STAT proteins [6]. Hence, regulation of JAK-STAT signaling could be a therapeutic target for dermatitis.

Keratinocytes, which make up 95% of skin epidermal cells, serve as the structural barrier of the epidermis and regulate the inflammatory and immunological responses of the skin [7]. HaCaT cells, a human keratinocyte cell line, are a model for studying anti-inflammatory and therapeutic effects in skin illnesses such as AD. The stimulation of interferon (IFN)-γ and tumor necrosis factor (TNF)-α in HaCaT cells causes the secretion of proinflammatory mediators, such as IL-1β, IL-6, IL-8, TARC, MDC, RANTES, and TNF-α [8].

Wild soybean (*Glycine soja* Sieb. & Zucc.; GS), a progenitor of cultivated soybean, is a traditional food native to east Asia, including China, Japan, eastern Russia, and Korea [9]. GS exhibits anti-oxidative, anti-aging, anti-hyperlipidemic, and anti-obesity properties [10,11]. In clinical studies of postmenopausal women, GS reduces the frequency of menopausal symptoms, such as hot flashes, via estrogenic effects [12]. GS contains a high content of flavonoids and has been used in cosmetics as a good emollient and moisturizer [13]. GS leaf and stem extract exerts anti-osteoarthritic properties by downregulating inflammation in a monosodium iodoacetate-induced osteoarthritis rat model and IL-1β-stimulated chondrocytes [14]. GS extract is invaluable for the development of functional food or medicine. Despite evidence of the benefits of GS extract in treating multiple conditions, no study has reported on the beneficial properties of GS extract in inflammatory skin disease. Therefore, we examined the effects of GS on skin inflammation in *Dermatophargoides farinae* extract (DfE)-induced AD mice and TNF-α/IFN-γ-stimulated keratinocytes. Because repeated contact with DfE allergens from NC/Nga mice causes clinical AD symptoms similar to humans with elevated circulating IgE levels [15], we used this model for our study.

## 2. Results

### 2.1. AD Symptoms

After the application of DfE or GS (Figure 1A), mice showed no differences in body weights between the groups (Figure 1B). Typical AD clinical symptoms were observed with the naked eye after the application of DfE to the skin of the mice. Oral GS administration resulted in the alleviation of AD-like lesions (Figure 1C). The dermatitis severity score, assessed 24 days after the first application of DfE, was increased in AD control mice, which are mice that received only DfE. In contrast, the dermatitis severity score was significantly decreased in GS- or dexamethasone-administrated mice (Figure 1D).

### 2.2. Effect of GS on Epidermal Thickness, Scratching, Transepidermal Water Loss (TEWL), and Spleen Weight

GS significantly reduced the ear thickness that was increased by DfE application (Figure 2A). TEWL decreased after the administration of GS (Figure 2B), suggesting that skin barrier disruptions were repaired. Scratching during 20 min was significantly decreased by GS treatment (Figure 2C). Weighing of the spleens to examine immune response revealed that the spleen weights were increased in only the DfE-treated control mice and decreased in the mice treated with GS (Figure 2D). Thus, our results suggest that GS alleviates AD in mice.

### 2.3. Effect of GS on AST, ALT, BUN, Creatinine, TSLP, and IgE Levels in Mouse Serum

The levels of AST and ALT, which are markers of liver injury, were not altered in GS-treated mice compared to only DfE-applied AD control mice (Figure 2E,F). The renal function markers, creatine and BUN, which were increased in AD mice, were reduced in mice treated with GS (Figure 2G,H). These results suggest that GS extract does not cause liver or kidney toxicity. Increased TSLP and total IgE levels in the mouse serum, which are indicators of AD, were remarkably reduced by GS treatment (Figure 3A,B). TSLP levels were reduced about 2.5-fold by GS administration, and IgE levels were recovered to normal levels in GS mice.

### 2.4. Skin Histopathology

To assess the effects of GS on AD mice, the ear skin epidermis thickness was examined in the hematoxylin and eosin (H&E)-stained sections (Figure 4A). The ear thickness and ear skin epidermis thickness increased in the AD skin lesions, and GS administration reduced the ear thickness and ear skin epidermis thickness (Figure 4B–D). Ear and skin sections were stained with toluidine blue and Congo red to examine the mast cells and eosinophils (Figure 5A,B). GS remarkably decreased the infiltration of these immune cells to the dermatitis sites (Figure 5C–F). These results suggest that GS induces the recovery of AD skin lesions by reducing dermal and epidermal thickness and reducing the infiltration of eosinophils and mast cells in the ear and dorsal skin.

### 2.5. AD-Related Gene Expression in the Skin

We further examined the effect of GS on the expression of AD-related cytokines. The mRNA expression levels of inflammatory cytokines (TARC, RANTES, MDC, and IL-6) were increased in the only DfE-applied AD control group compared to that in the untreated normal group. In contrast, the increased cytokine expression was inhibited in GS-treated mice (Figure 6A–D). Expression of filaggrin, a marker of skin barrier function, was decreased in the AD mice compared to that in the untreated normal mice, which received neither DfE, GS, nor dexamethasone. GS administration increased filaggrin expression compared to AD control mice (Figure 6E). Therefore, our results suggest that GS treatment inhibits DfE-induced AD via the inhibition of inflammatory cytokine expression and increased filaggrin expression.

### 2.6. Effect of GS on Signaling Pathways in the Skin

To determine the mechanism by which GS treatment in the AD mouse improves AD, we investigated the activation of the JAK-STAT pathway. In the JAK-STAT evaluation, lesional assessment focus on the local inflammation in the skin, while systemic assessment evaluates the broader immune response and body function. Thus, the phosphorylation levels of JAK and STAT proteins were investigated in the lesional skin. Phosphorylation of STAT1, STAT3, and JAK1 was elevated in the AD control mice compared to normal mice (Figure 7A–D and Appendix A). However, GS treatment significantly suppressed the phosphorylation of STAT1, STAT3, and JAK1 compared to AD mice that did not receive GS. In addition, filaggrin expression was upregulated in the GS-treated groups (Figure 7A–E). These findings suggest that GS treatment inhibits AD lesions by downregulating the JAK-STAT pathway in the DfE model of AD.

### 2.7. Quantitative Analysis of GS

Based on its absorption profile and retention time, GS contains 0.13 ± 0.004 mg/g genistein and 0.25 ± 0.017 mg/g apigenin. The mass spectrum data suggest that GS contains epicatechin ([M−H]^−^, 289.26), daidzin ([M+H]^+^, 417.44), genistin ([M+H]^+^, 433.42), daidzein ([M+H]^+^, 255.10), genistein ([M+H]^+^, 271.23), apigenin ([M+H]^+^, 271.24), and soyasaponin Bb ([M−H]^−^, 942.18; Figure 8).

### 2.8. Effect of GS and Its Components on RANTES Secretion in HaCaT Cells

The cytotoxicity of GS and its seven components was evaluated via cell viability in HaCaT cells. Neither GS nor its components showed toxicity in HaCaT cells (Figure 9A). In contrast, GS and all compounds except apigenin slightly increased cell viability. Next, the effect of GS and its components on RANTES secretion was assessed in HaCaT cells stimulated with TNF-α/IFN-γ. GS, apigenin, and genistein significantly suppressed the release of RANTES without cell toxicity (Figure 9B).

### 2.9. Effect of GS and Its Components on mRNA Expression and STAT Pathways in HaCaT Cells

Next, we studied the inhibitory properties of two compounds, apigenin and genistein, on the expression of other major inflammatory mediators. Apigenin and genistein reduced the expression of IL-6, TARC, and MDC mRNA, amplified filaggrin mRNA expression, and suppressed STAT3 phosphorylation in HaCaT cells (Figure 10 and Appendix A). These results suggest that apigenin and genistein increase the expression of inflammatory and AD-related genes via regulation of the STAT3 pathway.

## 3. Discussion

AD is characterized by skin barrier defects, inflammation, and persistent pruritus [16,17]. Skin barrier defects enhance permeability, allergen penetration, and Th2 differentiation by stimulating IL-4, expanding B cells for IgE class switching, and releasing inflammatory mediators to encourage allergic inflammation [18,19]. The loss of the late epidermal differentiation protein filaggrin plays a crucial role in skin barrier dysfunction in AD, leading to impaired formation of the impenetrable barrier known as the water loss (xerosis) and stratum corneum (ichthyosis) [20]. Increased serum levels of MDC and TARCs in AD patients correlate positively with IgE elevation, eosinophilia, and disease activity [21]. Several cytokines, including TSLP, IL-13, IL-31, and IL-4, are involved in pruritus, one of the most prominent features of AD [22]. Although it is difficult to replicate all human AD clinical features in a model, several AD in vivo models comprising spontaneous and induced and in vitro models, including human keratinocytes and 3D cells, have been used for designing pre-clinical studies for therapeutic discovery [23]. Keratinocytes are key players in the skin barrier function against environmental damage, skin immune systems, and initiating and pathogenesis of ski inflammation; thus, research using keratinocytes is very important for clinical understanding of AD [24].

In our study, GS extract improved symptoms in DfE-induced AD mice, such as scratching, increased epidermal thickness and TEWL due to skin barrier defects, penetration of immune cells (eosinophils and mast cells) to AD skin lesions, and elevation of IgE and TSLP in serum. In addition, MDC, RANTES, IL-6, and TARC were decreased by GS treatment in a dose-dependent manner in DfE-induced AD mice as well as IFN-γ/TNF-α-induced keratinocytes. In addition, GS treatment increased the expression of filaggrin in AD mice and keratinocytes. These results suggest that GS extract improves AD-like skin lesions through the downregulation of inflammatory mediators and upregulation of epidermis barrier function.

Activation of the JAK-STAT pathway plays a major part in the pathogenesis of inflammatory immune disorders, including AD, rheumatoid arthritis, psoriasis, and bowel disorders [25]. In AD, excessive activation of JAK1 induces the phosphorylation of STAT proteins, the hyperproliferation of keratinocytes, and the secretion of proinflammatory cytokines, leading to skin barrier disruption, the progression of pruritus and dermatitis symptoms, and pain [26]. Therefore, we studied the JAK-STAT signaling pathway to identify the mechanisms responsible for AD improvement by GS. DfE-induced hyperphosphorylation of JAK1, STAT1, and STAT3 was decreased in AD mice that received oral administration of GS. Although DfE decreased filaggrin expression in mice, GS administration upregulated the level of filaggrin compared to untreated AD mice. A previous study has reported that TSLP, a cytokine highly expressed by keratinocytes during the phase of AD, downregulates filaggrin expression and epidermal barrier function via the STAT3-dependent pathway [27]. These observations suggest that GS restores skin barrier damage via the upregulation of filaggrin expression and downregulation of JAK-STAT activation, thereby improving DfE-induced AD lesions. Similarly to our results, various traditional herbal medicines have been reported to improve AD by inhibiting the JAK-STAT pathway (Table 1) [16,28,29,30].

GS is known for its high flavonoid content [31]. In this study, six flavonoids and one terpenoid in the GS leaf and stem were detected, including flavones (apigenin), catechins (epicatechin), isoflavones (genistein, genistin, daidzin, and daidzein), and triterpenoid saponin (soyasaponin Bb). Among these major compounds, genistein and apigenin significantly decreased the gene expression of inflammatory mediators (IL-6, RNATES, TARC, and MDC) and increased filaggrin expression via the downregulation of the STAT3 pathway in IFN-γ/TNF-α-treated HaCaT cells. Our result is consistent with a previous report, which has identified therapeutic effects of genistein and apigenin in inflammatory skin disease. In the study, genistein suppressed the expression of inflammatory factors via the downregulation of STAT3 phosphorylation in imiquimod-induced psoriasis skin lesions in mice as well as in TNF-α-stimulated HaCaT cells [32]. The effects of genistein in AD have also been studied in conventionally housed NC/Nga mice, a severe, spontaneous AD model. In NC/Nga mice, genistein suppresses AD development via the inhibition of serum IgE levels and cytokine expression [33]. In rat basophilic leukemia cells, the murine macrophage cell line RAW264.7, and HaCaT keratinocytes, apigenin ameliorates allergic and inflammatory responses by inhibiting the production of cytokines (IL-6, TNF-α, and cyclooxygenase-2) and inducing skin barrier molecules (loricrin, filaggrin, and hyaluronic acid synthase) [34]. Apigenin also shows anti-inflammatory properties in psoriasis-like dermatitis in mice [35]. Moreover, topical apigenin treatment reduces inflammatory symptoms and TEWL in phorbol 12-myristate-, 13-acetate-, and oxazolone-treated allergic contract dermatitis animals [36]. Other studies have shown that apigenin inhibits IL-31 expression in the AD itch model mouse and activates mast cells via treatment with compound 48/80, which supports the potential of apigenin for treating AD [37]. These results suggest that genistein and apigenin may contribute to the anti-inflammatory and anti-atopic properties of GS in AD inflammatory skin disease.

The identified flavonoids in GS extract share structural similarities with isocoumarins, which have demonstrated potent anti-inflammatory, antioxidant, and neuroprotective activities via the leukotriene and prostaglandin pathways [38]. Therefore, additional research on these pathways may also be considered to study the mechanism of action of GS in AD.

There are various oxidative stress mechanisms involved in the pathogenesis of AD. Some flavonoids containing chlorogenic acid, wogonin, and quercetin have been shown to improve AD-related inflammation and oxidative stress by activating the antioxidant protective factors, such as nuclear factor erythroid 2-related factor 2 (Nrf2) and heme oxygenase-1 [39]. Thus, further studies of these compounds in DfE-induced AD models are needed to understand their anti-inflammatory effects and oxidative stress mechanisms in AD. Additionally, to understand the exact efficacy of GS, further studies are needed to elucidate potential synergistic effects between apigenin, genistein, and other GS components and how these compounds interact to amplify the observed effects.

A limitation of the current study is that only six mice per group were used. It needs to increase the sample size to enhance result reliability. Also, while JAK/STAT inhibition was demonstrated, the specific molecular targets or interactions with additional pathways such as NF-κB were not explored. The next study needs to evaluate additional mechanistic studies (e.g., gene knockout/overexpression) to pinpoint specific targets of GS with comparing GS’s efficacy to existing JAK inhibitors. Oral JAK inhibitors, such as baricitinib, upadacitinib, and abrocitinib, are used for patients with moderate to severe AD, offering a rapid reduction in burden and itch, and topical JAK inhibitors, such as ruxolitinib cream, are used for short-term treatment of mild to moderate AD [40,41]. This study does not address bioavailability or long-term safety of GS extract. Further studies are needed on the bioavailability of GS components, including aspects of absorption, metabolism, and the feasibility of topical application.

Another limitation of this study is that only seven flavonoids were identified and other potential active compounds were not investigated. Thus, it needs to expand component analysis to assess the bioactivity of other compounds in GS. Despite these limitations, the GS extract showed excellent AD improvement effects that were equal to or greater than current treatment (dexamethasone) without side effects such as renal or liver toxicity. Current AD treatments include topical corticosteroids, calcineurin inhibitors, and oral steroids focus on alleviating the AD symptoms but are usually accompanied by side effects, such as skin atrophy and drug resistance [42]. Therefore, GS could be a potential alternative therapeutic with minimized side effects of existing AD treatments.

## 4. Materials and Methods

### 4.1. Extraction of GS

GS leaves and stems were collected in the arenas of Munkyeong (Chungbuk, Republic of Korea). One hundred grams of GS were extracted with 70% ethanol for 3 h at reflux before being concentrated and freeze-dried. The extract yield was 12.3%.

### 4.2. Application of DfE and GS

To induce AD, Biostir AD ointment (Hyogo, Japan) was applied to the skin of the back and ear of NC/Nga mice twice a week. GS extract in doses of 30, 100, and 200 mg/kg was administered orally to the mice for 17 days. GS doses were determined from the prior dose-dependent experiments. Mice that received the anti-inflammatory drug dexamethasone served as positive controls. This study was permitted by the Animal Experimentation Ethics Committee of the Korea Institute of Oriental Medicine (approval number 22-078). A detailed description of the methods is presented in Appendix A and Figure 1A.

### 4.3. Assessment of Ear Thickness and Dermatitis Scores

Thickness of ear was determined by a micrometer. The dermatitis severity score, assessed by dryness, erythema, edema, and excoriation, was measured twice per week based on the following scale: 0, no signs; 1, mild; 2, moderate; 3, severe.

### 4.4. Determination of TSLP and IgE in Mouse Serum

TSLP and IgE concentrations in mouse serum were evaluated using TSLP kits (R&D, Minneapolis, MN, USA) and IgE kits (Shibayagi, Shibukawa, Gunma, Japan), respectively, according to the manufacturer’s manuals.

### 4.5. Skin Barrier Assessment

To assess the skin barrier function, transepidermal water loss (TEWL) was measured for 10 s with the gpskin Barrier Pro device (Gpower, Seoul, Republic of Korea), according to the manufacturer’s manuals. The barrier Pro device probe was in intimate contact with the mouse’s epidermis for about 10 s, and the measured value was recorded using the GPskin research app. (Version 2.5.5). Measurements were repeated three times, and the average TEWL score was used for analysis.

### 4.6. Evaluation of Scratching Behavior

Scratching was measured for 20 min, as described in previous methods [16]. The number of times each mouse scratched the skin of its neck, face, and back using its feet was measured visually for 20 min. Scratching behavior typically consisted of several scratching actions of about 1 s each. Any scratching behavior that lasted longer than 1 s was counted as one scratching session.

### 4.7. Measurement of Biomarkers for Liver and Kidney Function

Serum isolated from blood obtained from the inferior vena cava was used to measure aspartate aminotransferase (AST), blood urea nitrogen (BUN), alanine aminotransferase (ALT), and creatinine levels at Chaon, Inc. (Yongin, Republic of Korea).

### 4.8. Histopathological Analysis

Tissue samples (ear and back skin) were fixed with 10% formalin, embedded, sectioned at 5 μm, and stained with toluidine blue, hematoxylin and eosin (H&E), or Congo red. The epidermal thickness of the ear and back skin was evaluated 5 times along the length of each section.

### 4.9. Analytical Conditions

A detailed description of the methods is presented in Appendix A. The reference standards, apigenin, genistein, genistin, soyasaponin Bb, epicatechin, daidzein, and daidzin were obtained from Chemfaces (Wuhan, China).

### 4.10. HaCaT Cell Culture

The cells (4 × 10^4^ cells/well) were plated in 96-well microplates with Dulbecco’s Modified Eagle Medium (Gibco, Waltham, MA, USA), 10% fetal bovine serum, and 100 U/mL antibiotics. Then, the cells were treated with 10 ng/mL TNF-α/IFN-γ and either 10, 20, or 50 μg/mL GS extract for 24 h. Cell-free supernatants were collected, and the production of RANTES was determined using an enzyme-linked immunosorbent assay (ELISA) kit (R&D systems, Minneapolis, MN, USA). Cells were used for the isolation of either RNA or protein. To measure cell viability, cells were treated with cell counting kit-8 solution (Dojindo, Kumamoto, Japan) for 2 h. The values were determined at 450 nm absorbance.

### 4.11. Reverse Transcription–Quantitative Polymerase Chain Reaction (RT-qPCR)

Total RNA from mouse skin tissue or cells was extracted, and the gene expression from synthesized cDNA was determined using the RT-qPCR. A detailed description of the methods is presented in Appendix A.

### 4.12. Immunoblot

Proteins extracted from skin tissue or cells were electrophoresed, transferred, and incubated with primary antibodies. A detailed description of the methods is presented in Appendix A.

### 4.13. Statistical Analysis

All values are presented as the mean ± standard error of the mean (SEM; for animal studies) or standard deviation (SD; for cell studies). Statistical significance was determined via a one-way ANOVA (GraphPad Prism v7.05) with Dunnett’s multiple test: # or *, *p* < 0.05; ## or **, *p* < 0.01; ### or ***, *p* < 0.001.

## 5. Conclusions

This study shows that GS extract ameliorated skin inflammation via regulation of the JAK-STAT signaling pathway in an AD murine model and keratinocytes. Our findings provide new evidence that GS leaf and stem extract may be a therapeutic candidate for AD.

## Figures and Tables

**Figure 1 ijms-26-04560-f001:**
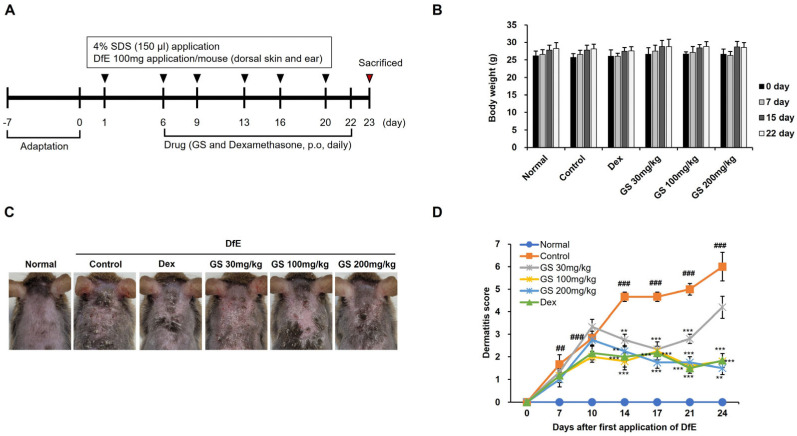
Effect of *Glycine soja* (GS) on atopic dermatitis (AD) skin lesions in *Dermatophargoides farinae* extract (DfE)-induced AD mice. (**A**) Experimental design for the in vivo study using a DfE-induced AD model. (**B**) Body weight. (**C**) Representative picture of DfE-induced AD mice. (**D**) Dermatitis score. Mice were separated into six groups: Normal (untreated), Control (treated with DfE only), 30 mg/kg GS (treated with DfE), 100 mg/kg GS (treated with DfE), 200 mg/kg GS (treated with DfE), and Dex (Dexamethasone, positive control). Results are presented as the mean ± standard error of the mean (SEM; n = 6). ##, *p* < 0.01 vs. Normal; ###, *p* < 0.001 vs. Normal; **, *p* < 0.01 vs. Control; ***, *p* < 0.001 vs. Control.

**Figure 2 ijms-26-04560-f002:**
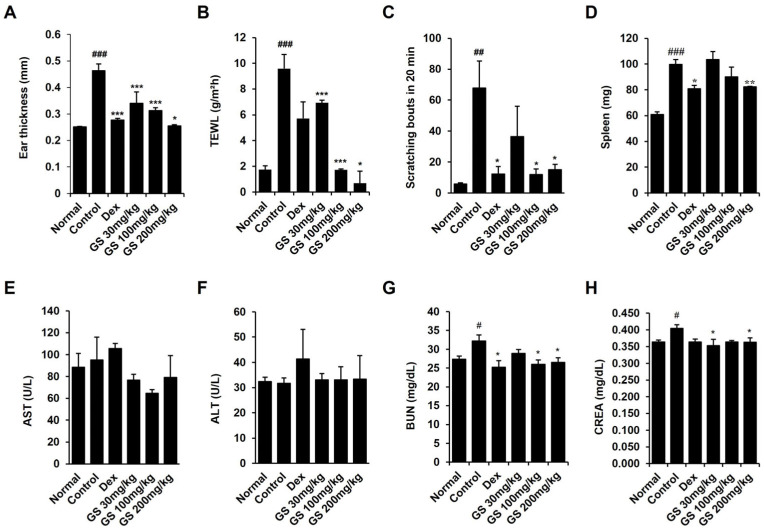
Effect of GS on ear thickness, transepidermal water loss (TEWL), scratching, spleen weight, and serum aspartate aminotransferase (AST), alanine aminotransferase (ALT), blood urea nitrogen (BUN), and creatine in AD mice. (**A**) Ear thickness. (**B**) TEWL. (**C**) Number of scratching behaviors. (**D**) Spleen weight. Measurement of (**E**) AST, (**F**) ALT, (**G**) BUN, and (**H**) creatine levels in serum samples from mice. Results are presented as the mean ± SEM (n = 6). #, *p* < 0.05 vs. Normal; ##, *p* < 0.01 vs. Normal; ###, *p* < 0.001 vs. Normal; *, *p* < 0.05 vs. Control; **, *p* < 0.01 vs. Control; ***, *p* < 0.001 vs. Control.

**Figure 3 ijms-26-04560-f003:**
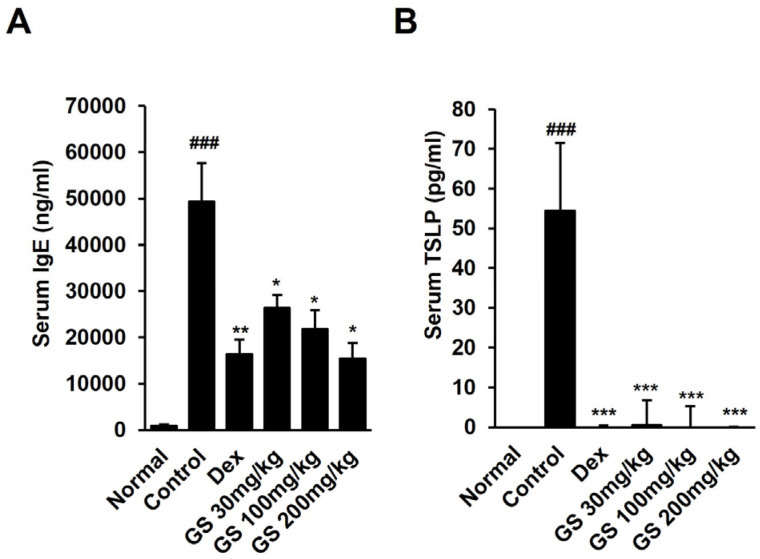
Effect of GS on IgE and thymic stromal lymphopoietin (TSLP) in mouse serum. (**A**) Total IgE and (**B**) TSLP levels in serum were measured using ELISA. Results are presented as the mean ± SEM (n = 6). ###, *p* < 0.001 vs. Normal; *, *p* < 0.05 vs. Control; **, *p* < 0.01 vs. Control; ***, *p* < 0.001 vs. Control.

**Figure 4 ijms-26-04560-f004:**
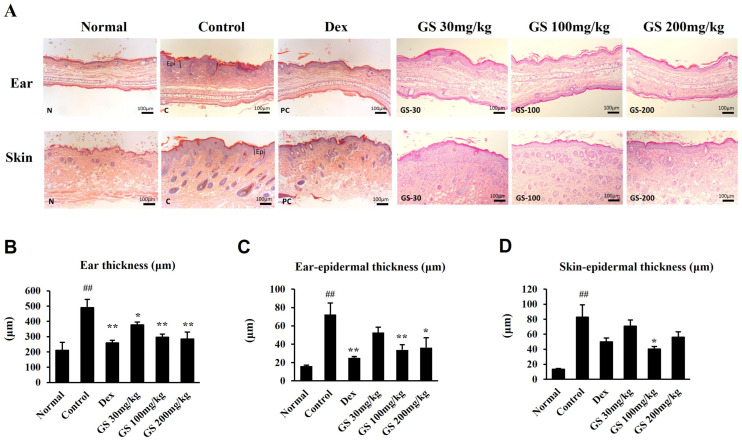
Effect of GS on ear and dorsal skin thickness. (**A**) Hematoxylin and eosin staining of ear and dorsal skin. (**B**) Ear thickness. (**C**) Ear epidermal thickness. (**D**) Dorsal skin epidermal thickness. Stained tissue was photographed under a light microscope at 100× magnification (scale bar: 100 μm). Results are presented as the mean ± SEM (n = 6). ##, *p* < 0.01 vs. Normal; *, *p* < 0.05 vs. Control; **, *p* < 0.01 vs. Control.

**Figure 5 ijms-26-04560-f005:**
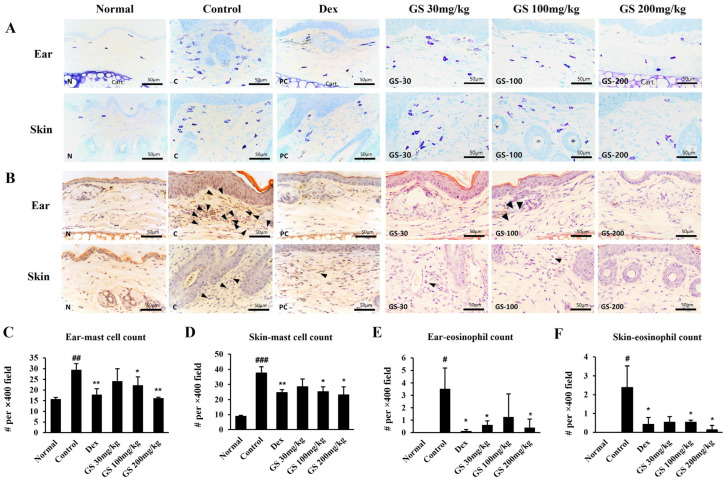
Effect of GS on the recruitment of mast cells and eosinophils in AD lesions. Images of (**A**) toluidine blue and (**B**) Congo red staining of ear and dorsal skin. Number of (**C**,**D**) mast cells and (**E**,**F**) eosinophils. Stained sections were photographed under a light microscope at 400× magnification (scale bar, 50 μm). Results are presented as the mean ± SEM (n = 6). #, *p* < 0.05 vs. Normal; ##, *p* < 0.01 vs. Normal; ###, *p* < 0.001 vs. Normal; *, *p* < 0.05 vs. Control; **, *p* < 0.01 vs. Control. Triangle, eosinophils.

**Figure 6 ijms-26-04560-f006:**
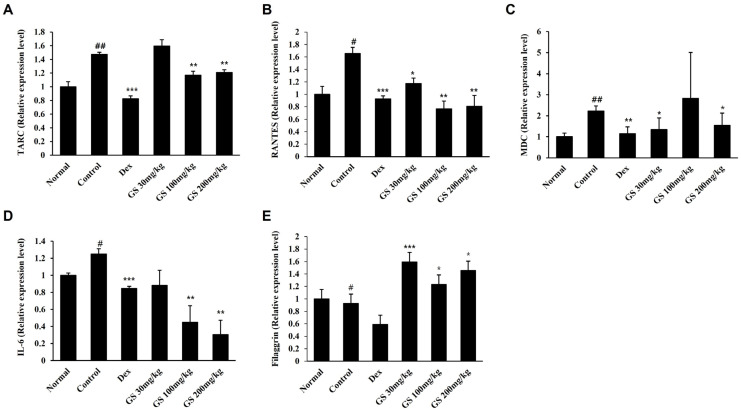
Effect of GS on mRNA expression of inflammatory cytokines and filaggrin in AD mice. The mRNA levels of (**A**) thymus- and activation-regulated chemokine (TARC), (**B**) regulated on activation, normal T cell expressed and secreted (RANTES), (**C**) macrophage-derived chemokine (MDC), (**D**) interleukin (IL)-6, and (**E**) filaggrin in AD skin lesions were evaluated via reverse transcription–quantitative polymerase chain reaction (qRT-PCR). Results are presented as the mean ± SEM (n = 3). #, *p* < 0.05 vs. Normal; ##, *p* < 0.01 vs. Normal; *, *p* < 0.05 vs. Control, **, *p* < 0.01 vs. Control; ***, *p* < 0.001 vs. Control.

**Figure 7 ijms-26-04560-f007:**
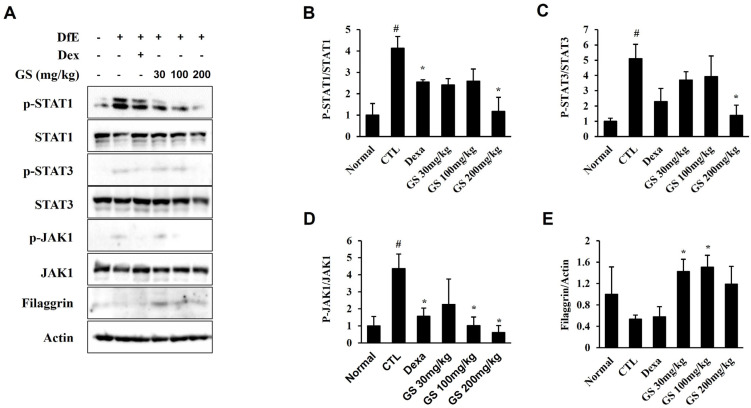
Effect of GS on Janus kinase 1 (JAK1)–signal transducer and activator of transcription (STAT) 1/3 pathway activation in AD skin lesions. (**A**) Immunoblotting was performed for p-STAT1, STAT1, p-STAT3, STAT3, p-JAK1, JAK1, and filaggrin. (**B**–**D**) Phosphorylated protein levels were quantified as the density relative to the expression of total protein. (**E**) Filaggrin levels were quantified as the density relative to the expression of actin. Results are presented as the mean ± SEM (n = 3). #, *p* < 0.05 vs. Normal; *, *p* < 0.05 vs. Control.

**Figure 8 ijms-26-04560-f008:**
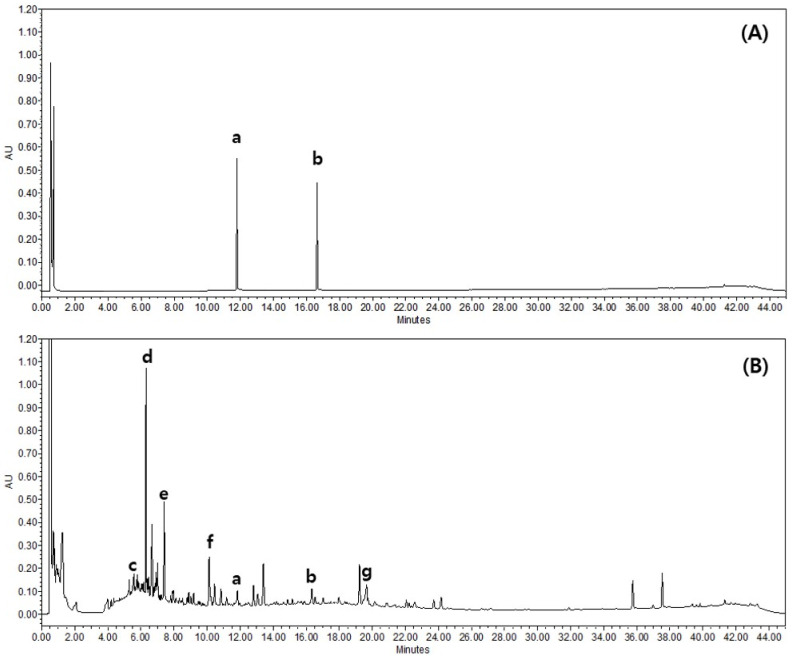
Representative ultra-performance liquid chromatography (UPLC) chromatogram of 70% ethanol extract of GS leaf and stem. (**A**) Mixed standards solution; (**B**) 70% ethanol extract of GS, (a) genistein, (b) apigenin, (c) epicatechin, (d) daidzin, (e) genistin, (f) daidzein, (g) soyasaponin Bb.

**Figure 9 ijms-26-04560-f009:**
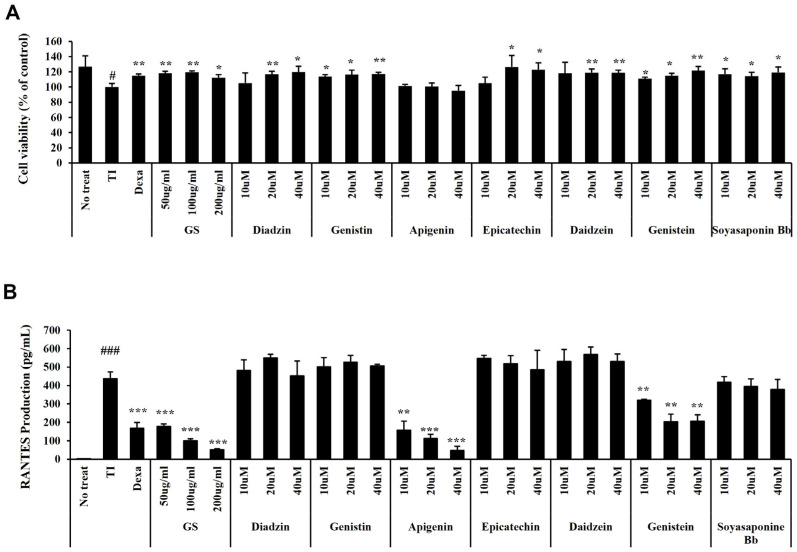
Effect of GS constituents on cell viability and RANTES production in HaCaT cells. (**A**) Cell viability using the cell counting kit (CCK)-8 assay. (**B**) RANTES production by enzyme-linked immunosorbent assay (ELISA). Results are presented as the mean ± standard deviation (SD) of three independent experiments. #, *p* < 0.05 vs. No treatment; ###, *p* < 0.001 vs. No treatment; *, *p* < 0.05 vs. Tumor necrosis factor (TNF)-α/interferon (IFN)-γ-treated control; **, *p* < 0.01 vs. TNF-α/IFN-γ-treated control; ***, *p* < 0.001 vs. TNF-α/IFN-γ-treated control.

**Figure 10 ijms-26-04560-f010:**
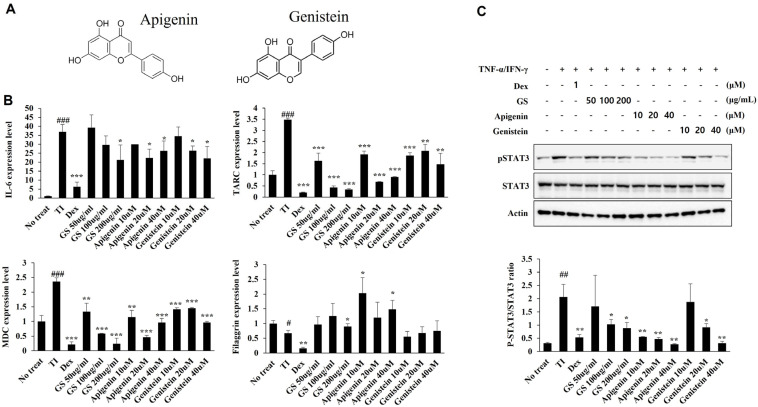
Effect of apigenin and genistein on mRNA expression of inflammatory cytokines and filaggrin and STAT3 phosphorylation in HaCaT cells. (**A**) Chemical structure of apigenin and genistein. (**B**) Expression of TARC, MDC, IL-6, and filaggrin mRNA. (**C**) STAT3 phosphorylation. Results are presented as the mean ± SD of three independent experiments. #, *p* < 0.05 vs. No treatment; ##, *p* < 0.01 vs. No treatment; ###, *p* < 0.001 vs. No treatment; *, *p* < 0.05 vs. TNF-α/ IFN-γ-treated control; **, *p* < 0.01 vs. TNF-α/IFN-γ-treated control; ***, *p* < 0.001 vs. TNF-α/ IFN-γ-treated control.

**Table 1 ijms-26-04560-t001:** Traditional herbal medicines affecting JAK-STAT pathway in AD.

Traditional Medicine	Application	Reference
*Reynoutria japonica*	Oral administraton	[28]
*Evodiae Fructus*	Topical application	[29]
*Pulsatilla koreana*	Oral administration	[30]
*Securinega suffruticosa*	Oral administration	[16]

## Data Availability

Data will be made available on request.

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
