# Peer review of "Glycine soja* Leaf and Stem Extract Ameliorates Atopic Dermatitis-like Skin Inflammation by Inhibiting JAK/STAT Signaling"

_ijms, 2025, doi:10.3390/ijms26104560_

Round 1
Reviewer 1 Report
Comments and Suggestions for Authors
The article titled “Glycine soja leaf and stem extract ameliorates atopic dermatitis-like skin inflammation by inhibiting JAK/STAT signaling” investigates the therapeutic potential of Glycine soja (GS) leaf and stem extract in alleviating atopic dermatitis (AD)-like symptoms via JAK/STAT pathway inhibition. Using a Dermatophagoides farinae extract-induced mouse model and TNF-α/IFN-γ-stimulated keratinocytes, the authors demonstrate that oral GS administration reduces scratching, epidermal thickening, inflammatory markers (IL-6, TARC, MDC, RANTES), and serum IgE/TSLP while upregulating filaggrin. Mechanistically, GS suppresses JAK1, STAT1, and STAT3 phosphorylation. UPLC analysis identifies seven constituents, including apigenin and genistein, which replicate GS’s anti-inflammatory effects in vitro. The study is methodologically robust, integrating in vivo and in vitro models, histopathology, and molecular profiling. However, the classification of soyasaponin Bb as a flavonoid is inconsistent with standard biochemical definitions, creating ambiguity. The dose-dependent effects of GS in mice lack clear justification for the selected concentrations (30/100/200 mg/kg). While the discussion contextualizes GS’s role in JAK/STAT inhibition, it overlooks comparisons to existing JAK inhibitors (e.g., baricitinib) and potential synergies between GS constituents. The study also does not address bioavailability or long-term safety of GS extract, which are critical for translational relevance. Additionally, the in vitro experiments focus narrowly on STAT3, neglecting other pathways (e.g., NF-κB) that may contribute to AD pathogenesis. Despite these gaps, the findings provide valuable preclinical evidence for GS as a functional food or topical agent for AD. The reviewer has the following comments that authors need to address.
- Reclassifying soyasaponin Bb as a triterpenoid saponin rather than a flavonoid would improve the accuracy of compound classification and enhance the clarity of the phytochemical analysis.
- Justifying the selection of GS doses (30, 100, and 200 mg/kg) in mice would strengthen the study design by clarifying whether the chosen concentrations are based on pharmacological relevance, prior studies, or toxicity thresholds.
- Expanding the mechanistic scope by investigating additional pathways such as NF-κB and comparing GS’s efficacy to existing JAK inhibitors would provide a more comprehensive understanding of its therapeutic potential. Additionally, addressing the bioavailability of GS constituents including aspects of absorption, metabolism, and the feasibility of topical application would enhance the translational relevance of the findings.
- Refining statistical reporting by differentiating the use of SEM and SD (e.g., for animal versus cell studies) in the figure legends would improve clarity and ensure accurate interpretation. Additionally, elucidating potential synergies between apigenin, genistein, and other GS constituents could provide valuable insights into how these compounds interact to amplify the observed effects.
- The identified flavonoids in GS extract share structural similarities with isocoumarins, which have demonstrated potent anti-inflammatory, antioxidant and neuroprotective activities. The authors are encouraged to cite the following related isocoumarin based studies to strengthen the discussion on small-molecule modulation of inflammatory signaling.
https://www.sciencedirect.com/science/article/abs/pii/S0223523416307243
- Strengthening clinical relevance by including data from human keratinocytes or ex vivo skin models would help bridge the gap between preclinical findings and potential therapeutic applications. Additionally, discussing limitations such as the absence of long-term toxicity data and the translational challenges associated with oral GS administration would provide a more balanced and informative perspective.

Reviewer 2 Report
Comments and Suggestions for Authors
nice presented absract
in intro present clearly Jak-Stat pathway in atopic dermatitis- add a figure
the results ARE are well presented
difference of Jak-stat assessment in lesional and systemic evaluations in atopic dermatitis would be highlighted
report a table with other traditional medicine ( such as other flavonoids )affecting this pathway - and the way of application ( topical and per os)
can the flavonoid affect atopic dermatitis with other ways such as oxidative stress? report in discussion
report more in detail certain methods section suchas TEWL, SCRACHING MEASURES eTC
Reviewer 3 Report
Comments and Suggestions for Authors
This study investigates the therapeutic effects of wild soybean (Glycine soja, GS) leaf and stem extract on atopic dermatitis (AD)-like skin inflammation and its underlying mechanisms. Using a Dermatophagoides farinae extract (DfE)-induced AD mouse model and HaCaT keratinocytes, the authors demonstrated that GS extract alleviated AD symptoms, including scratching, epidermal thickening, and inflammatory cell infiltration, while reducing serum levels of TSLP and IgE. GS downregulated inflammatory cytokines (e.g., IL-6, RANTES, TARC, MDC) and upregulated filaggrin expression by inhibiting the JAK/STAT signaling pathway. Seven flavonoids in GS were identified via UPLC, with apigenin and genistein validated for their anti-inflammatory effects.
1. Only 6 mice per group were used, which may limit the robustness and generalizability of the findings.
2. While JAK/STAT inhibition was demonstrated, the specific molecular targets or interactions with other pathways were not explored.
3. Only 7 flavonoids were identified; other potential active compounds were not investigated.
4. The discussion does not sufficiently compare GS with existing AD treatments to highlight its potential advantages or limitations.
Recommendations:
1. Increase the sample size to enhance result reliability.
2. Include additional mechanistic studies (e.g., gene knockout/overexpression) to pinpoint specific targets of GS.
3. Expand component analysis to assess the bioactivity of other compounds in GS.
4. Enrich the discussion with comparisons to current therapies to emphasize the unique potential or limitations of GS.
Overall, the study is well-designed and provides valuable insights into the anti-AD effects of GS. Addressing the above limitations could further strengthen its impact.
Round 2
Reviewer 1 Report
Comments and Suggestions for Authors
The authors have thoroughly addressed the comments raised by the previous reviewers, demonstrating careful attention to detail and a commitment to improving the manuscript. In my opinion, the article, in its current form, meets the standards of quality and scientific rigor expected for publication in the IJMS journal.
Reviewer 2 Report
Comments and Suggestions for Authors
the authors did take my suggestions into account and the manuscript was improved